# Performance Comparison of LMNO Cathodes Produced with Pullulan or PEDOT:PSS Water-Processable Binders

Alessandro Brilloni [1], Francesco Marchesini [1], Federico Poli [1], Elisabetta Petri [1] and Francesca Soavi [1,2,3,*]

1   Department of Chemistry "Giacomo Ciamician", Alma Mater Studiorum University of Bologna, Via Selmi 2, 40126 Bologna, Italy; alessandro.brilloni2@unibo.it (A.B.); francesco.marchesini@studio.unibo.it (F.M.); federico.poli8@unibo.it (F.P.); elisabetta.petri2@unibo.it (E.P.)
2   ENERCube, Centro Ricerche Energia, Ambiente e Mare, Centro Interdipartimentale per la Ricerca Industriale Fonti Rinnovabili, Ambiente, Mare ed Energia (CIRI-FRAME), Alma Mater Studiorum University of Bologna, Viale Ciro Menotti 48, 48122 Marina di Ravenna, Italy
3   National Reference Center for Electrochemical Energy Storage (GISEL)-INSTM, Via G. Giusti 9, 50121 Firenze, Italy
*   Correspondence: francesca.soavi@unibo.it

**Abstract:** The aim of this paper is to demonstrate lithium metal battery cells assembled with high potential cathodes produced by sustainable processes. Specifically, $LiNi_{0.5}Mn_{1.5}O_4$ (LMNO) electrodes were fabricated using two different water-processable binders: pullulan (PU) or the bifunctional electronically conductive poly(3,4-ethylenedioxythiophene)-poly(styrene sulfonate) (PEDOT:PSS). The cell performance was evaluated by voltammetric and galvanostatic charge/discharge cycles at different C-rates with 1M $LiPF_6$ in 1:1 (*v:v*) ethylene carbonate (EC):dimethyl carbonate (DMC) (LP30) electrolyte and compared to that of cells assembled with LMNO featuring poly(vinylidene difluoride) (PVdF). At C/10, the specific capacity of LMNO-PEDOT:PSS and LMNO-PU were, respectively, 130 mAh $g^{-1}$ and 127 mAh $g^{-1}$, slightly higher than that of LMNO-PVdF (124 mAh $g^{-1}$). While the capacity retention at higher C-rates and under repeated cycling of LMNO-PU and LMNO-PVdF electrodes was similar, LMNO-PEDOT:PSS featured superior performance. Indeed, lithium metal cells assembled with PEDOT:PSS featured a capacity retention of 100% over 200 cycles carried out at C/1 and with a high cut-off voltage of 5 V. Overall, this work demonstrates that both the water-processable binders are a valuable alternative to PVdF. In addition, the use of PEDOT:PSS significantly improves the cycle life of the cell, even when high-voltage cathodes are used, therefore demonstrating the feasibility of the production of a green lithium metal battery that can exhibit a specific energy of 400 Wh $kg^{-1}$, evaluated at the electrode material level. Our work further demonstrates the importance of the use of functional binders in electrode manufacturing.

**Keywords:** lithium-ion battery cathode; LMNO; pullulan; PEDOT:PSS; water processable binder; electronically conducting polymer

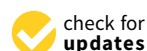



## 1. Introduction

The transition towards the use of cobalt-free, high voltage, and capacity cathode active materials (CAM), such as Li $(Ni_xMn_y)O_2$ (LMNO), will bring about the so-called cell generation Gen3b [1]. $LiNi_{0.5}Mn_{1.5}O_4$ is one of the most promising cathode formulations, featuring a theoretical specific capacity of 145 mAh $g^{-1}$ with working potential of ca. 4.7 V vs. $Li^+/Li$ [2].

While it displays good electronic and $Li^+$ conductivities, and excellent rate capability, it may exhibit severe capacity fade over cycling, especially at elevated temperatures [3,4]. To address the European Battery Alliance (EBA) goal [5], a valuable practical strategy that is being pursued is the shifting of the electrode productive process from organic solvents through to aqueous ones exploiting bio-derived and water processable binders [6].

Nowadays, poly(vinylidene difluoride) (PVdF) is commonly used for its good chemical and electrochemical stability, and solubilized N-methyl-2-pyrrolidone (NMP). NMP has been listed by Europe as a high concern substance [6,7]. It is toxic and requires expensive atmosphere-controlled environments, affecting the economic and environmental impact on cathode and Lithium Batteries (LIBs) manufacturing. Indeed, it has been reported that electrode drying and NMP recovery process imposes an energy demand of ~10 kWh per kg of NMP as it evaporates at 200 °C [6,8].

Commonly used aqueous binders are less expensive than PVdF and require a lower drying temperature and time [6]. Furthermore, for their low environmental impact during the manufacturing and the end-of-life management, they open new approaches towards the design for recycling of LIBs [9,10]. Water-processable polymers, such as cellulose, alginate, guar gum, have already been reported as successful alternatives to PVdF for CAMs [6,8–13].

Pullulan, for its high water-solubility, excellent mechanical properties, $O_2$-impermeability, stability at high pH, and biodegradability, is emerging as a green electrode binder. Pullulan has been reported as a binder component of silicon anodes, as well as pullulan: glycerol 1:1 mixture (PU) exhibited excellent binding capability for thick carbon electrodes of supercapacitors and $Li(Ni_{0.5}Mn_{0.3}Co_{0.2})O_2$ (NMC532) cathodes [10,13–15]. Supercapacitors working with PU-based electrodes and separators, combined with ionic liquid electrolyte, were demonstrated to be stable over thousands of cycles operated with a cell voltage of 3.2 V [14,15]. In [10], we demonstrated PU-based NMC532 featuring up to 167 mAh $g^{-1}$ of NMC532, and excellent cycling stability over 500 cycles. On the other hand, electronically conductive polymers conjugate the typical characteristics of polymeric materials, such as the high chemical inertia, good mechanical properties, and lightness with the ability to conduct electrons. Among the conducting polymers, poly(3,4-ethylenedioxythiophene) (PEDOT) is one of the most popular and it has been widely proposed as a key component for the next-generation consumer electronics and energy storage devices [16,17]. PEDOT has been already demonstrated as a valuable component of cathodes in lithium-ion batteries. Ozerova et al. showed that mechanical mixing of $LiFePO_4/C$ with pre-polymerized PEDOT particles, in the presence of different surfactants (Triton X-100 and cetyltrimethylammonium bromide) improved the cathode specific capacity by 20% with respect to the uncoated $LiFePO_4$, reaching 81 mAh $g^{-1}$ at 1600 mA $g^{-1}$ [18]. In situ oxidative polymerization of 3,4-ethylenedioxythiophene (EDOT) in acidic media has been used by Jinfeng Liu et al. to coat particles of LMNO with PEDOT. The resulting material has been exploited to prepare electrodes with a PVdF-binder. PEDOT-coated LMNO particles featured a better capacity retention with respect to the pristine material, with improved gravimetric capacity [19]. Laisuo et al. showed that a protective thin film of PEDOT on LMNO obtained by chemical vapor deposition (CVD) increases the rate capacity and extends the high temperature (50 °C) cycle life of $LiMn_2O_4$ by over 60% [20]. Moreover, by operando synchrotron X-ray diffraction, they demonstrated that PEDOT further improves current homogeneity in $LiCoO_2$ electrodes during cycling [21].

PEDOT doped with poly(styrene sulfonate) (PSS) anion (PEDOT:PSS) features high conductivity (10–$10^2$ S cm$^{-1}$), and nowadays commercial aqueous solutions of PEDOT:PSS are available. PEDOT:PSS has been exploited to coat particles that have been used to prepare electrodes with PVdF binder. PEDOT:PSS-coated $Li_{1.2}Ni_{0.2}Mn_{0.6}O_2$ featured 286 mAh $g^{-1}$ at 0.1 C and 146 mAh $g^{-1}$ at C/1 after 100 cycles [22]. Electrospun of LMNO nanoparticles coated with 1%wt of PEDOT:PSS exhibited improved cycle stability as side reactions with the electrolyte were alleviated. By this approach, up to 128 mAh $g^{-1}$ were achieved, but cycling stability was demonstrated only over 30 cycles [23].

The availability of commercial, aqueous PEDOT:PSS solutions opens the possibility to use it both as CAM coating and as a "bifunctional", green, electronically conductive binder. Along with its binding ability, PEDOT:PSS provides conductive bridges between individual particles, which can improve the electron transport within the electrode components [24]. Furthermore, PEDOT:PSS is attracting much interest as self-healing material, i.e., as a material that is able to recover its functionalities after being damaged. This property could

be seen as an additional advantage that might positively impact on the cathode cycling stability and LIB safety [25]. The good mechanical properties of PEDOT:PSS binder have also been demonstrated with silicon-based anodes, which are known for their relevant volume expansion/contraction during the lithiation/delithiation processes [26–28].

While PEDOT:PSS binder has been successfully demonstrated for LiFePO$_4$, LiCoO$_2$, and NMC composite electrodes [17,29], to the best of our knowledge, PEDOT:PSS has not been reported as bifunctional binder for high-voltage LMNO electrodes, yet.

The aim of this work is to demonstrate high-voltage lithium metal battery cells, assembled with a cathode produced by sustainable processes. Specifically, LMNO electrodes were fabricated using two different water-processable binders: pullulan (PU) or the bifunctional electronically conductive poly(3,4-ethylenedioxythiophene)-poly(styrene sulfonate) (PEDOT:PSS). While PU and PEDOT:PSS feature different structure and chemical–physical properties, they share three important features: they both (i) have high technological and market relevance, (ii) can be aqueous processed, and (iii) are biocompatible. Pullulan is indeed attracting much attention in the packaging industry, mainly in pharmaceutics and food [30,31]. In turn, PEDOT:PSS has been demonstrated to be a key material for organic electronics, sensing, and wearable applications [16]. In these fields, processability and technological exploitation of these polymers have been widely demonstrated. With our study we aim to further demonstrate the use of these two polymers in an additional, strategic technological sector, namely, LIB manufacturing.

The performance of the cells assembled with the two different aqueous binders was evaluated by voltammetric and galvanostatic charge/discharge cycles at different C-rates with 1M LiPF$_6$ in 1:1 (*v:v*) ethylene carbonate (EC):dimethyl carbonate (DMC) (LP30 electrolyte) and compared to that of cells assembled with LMNO featuring poly(vinylidene difluoride) (PVdF). The results demonstrated that PU and PEDOT:PSS are valuable green water processable binder alternatives to PVdF for high-voltage cathode that can guarantee high performance and, in the case of PEDOT:PSS, enhance the cycling stability.

## 2. Materials and Methods

### 2.1. Preparation of the Electrodes

Electrodes were prepared using the commercial LMNO powder, purchased from NANOMYTE® SP-10 (NEI Corporation, Somerset, NJ, USA), as active material. The powder stoichiometry is LiMn$_{1.5}$Ni$_{0.5}$O$_4$, the average particle size is 4–7 μm, and the nominal capacity (at C/10, between 3.5 and 5 V vs. Li$^+$/Li) is 125 mAh g$^{-1}$. Different cathodes formulations have been studied. The identification codes and the areal mass loadings of the composites and LMNO of the electrodes produced by different binders are reported in Table 1. In LMNO-PU electrodes, the utilization of the bio derived, water processable binder pullulan:glycerol 1:1 in weight (PU) was investigated. They featured 85% LMNO, 10% carbon conductive additive (Super C45, Imerys, Paris, France), and 5% PU (2.5% pullulan and 2.5% glycerol). LMNO-PU composite mass loading was 2.1 mg cm$^{-2}$. The effect of the use of the bifunctional binder PEDOT:PSS was explored with LMNO-PE electrodes that featured 85% LMNO, 10% conductive carbon (Super C45), and 5% PEDOT:PSS. LMNO-PE composite mass loading was 3.8 mg cm$^{-2}$. As a benchmark, electrodes were also produced with PVdF binder and processed with NMP solvent, and they are labelled in the following text as LMNO-PVdF. Their composition was 85% LMNO, 10% conductive carbon (Super C65, Imerys), and 5% PVdF, the composite mass loading was 3.7 mg cm$^{-2}$. For LMNO-PU production, at first, pullulan powder (TCI) was dissolved in Milli-Q water with Glycerol (GLY, Sigma-Aldrich, Merk Life Science S.r.l., Milan, Italy) with a mass ratio 1:1 and stirred for 30 min. LMNO and carbon conductive additive (Super C45), with a mass ration 85:10 was added in a jar with the binder obtaining a slurry that was milled at 250 rpm for 30 + 30 (reverse) min in a planetary mill (FRITSCH, Pulverisette, Lainate, Milan, Italy). The slurry was subsequently casted on aluminum foil by a Mini Coating Machine (Hohsen Corporation, Osaka, Japan) at 0.3 cm s$^{-1}$ and with a bar distance of 8 mil (250 μm). The coated films were dried at 60 °C overnight in a thermostatic oven, pressed at

4 ton cm$^{-2}$, and dried again under dynamic vacuum (Büchi glass oven B-585) overnight at 60 °C to eliminate any water trace. The LMNO-PE electrodes were prepared by mixing the LMNO powder and the conductive carbon, with a mass ratio of 85:10 by dry milling at 250 rpm for 5 min in the planetary mill. Then, the aqueous solution of PEDOT:PSS (Sigma-Aldrich, 1.1%) was added to the jar, resulting in a slurry that was milled at 250 rpm for 1 h (30 min reverse). The slurry was subsequently casted on aluminum foil, dried, and pressed following the same procedure used for LMNO-PU electrodes. For the LMNO-PVdF production, LMNO and the conductive carbon powders with a mass ratio of 85:10, were dry milled at 250 rpm for 5 min. Then, a solution of PVdF in NMP was added to the jar, resulting in a slurry that was milled at 250 rpm for 1 h (30 min reverse). The slurry was subsequently casted on aluminum foil, dried at 60 °C overnight in a thermostatic oven, pressed, and dried again under dynamic vacuum at 120 °C to eliminate any solvent trace.

Finally, all the prepared electrodes were transferred and stored in a dry box under Argon atmosphere (MBraun, $H_2O$ and $O_2$ < 1 ppm).

**Table 1.** Acronyms, and composite and LMNO areal loadings of the cathodes produced with pullulan or PEDOT:PSS or PVdF binders.

| Electrode Name | Binder | Composite Mass Loading (mg cm$^{-2}$) | LMNO Mass Loading (mg cm$^{-2}$) |
| --- | --- | --- | --- |
| LMNO-PU | Pullulan | 2.1 | 1.8 |
| LMNO-PE | PEDOT:PSS | 3.8 | 3.2 |
| LMNO-PVdF | PVdF | 3.7 | 3.2 |

### 2.2. Chemical–Physical Analyses

X-ray diffraction (XRD) spectra were collected by a PANalytical X'Pert PRO powder diffractometer equipped with a X'Celerator detector (CuKα radiation, λ = 1.5406 Å, 40 mA, 40 kV), radiation source, and Ni filter by continuous scanning mode (step 0.017° 2θ step size, 10 s/step scan rate). Thermogravimetric Analysis (TGA) were performed by a TA Thermal Analysis Q50 equipment, under $O_2$ flow (60 mL min$^{-1}$) with a scan rate of 10 °C min$^{-1}$ up to 550 °C.

### 2.3. Lithium Metal/LMNO Coin Cell Prototype Assembly

The prepared electrodes were cut into 18 mm disks and tested as cathode of 2032 coin cells 2032 featuring 200 μL of a 1M LiPF$_6$ in 1:1 (*v:v*) ethylene carbonate (EC): dimethyl carbonate (DMC) (LP30, Selectilyte BASF, Ludwigshafen, Germany) solution as electrolyte with commercial Celgard 2300 as separator, and metallic lithium 16 mm as quasi-reference counter electrode. All the tested cells were assembled in a dry box (Labmaster 130, $H_2O$, and $O_2$ < 0.1 ppm MBraun, Garching, Germany).

### 2.4. Electrochemical Characterization

The electrochemical test consisted in cyclic voltammetry (CV) and galvanostatic charge/discharge cycles with potential limitation (GCPL). The electrochemical tests were performed in a thermostatic oven at 30 °C with a BioLogic VSP multichannel potentiostat/galvanostat/FRA. The measurements were carried with a two-electrode set up, in which the working electrode was the tested material, and the counter-reference was the lithium disk (see Section 2.3). All the Li/LMNO cells were tested with charge and discharge cell cut-off voltages of 5 V and 3.5 V, respectively. In particular, the test protocol consisted of a preliminary galvanostatic charge/discharge cycles at C/10 (conditioning cycles), carried out to form a stable CEI (Cathode Electrolyte Interface), required to let the electrode working above 4 V. The C/rate currents were set referring to the experimental specific capacity declared by the LMNO manufacturer, i.e., 125 mAh g$^{-1}$. This step was followed by 3 cyclic voltammetries (CVs) at 50 μV s$^{-1}$ to highlight any side electrochemical signals that could impact on cycling performance of the cell. CV discharge curves were analyzed to obtain a first evaluation of the specific capacity that was calculated from the integral of

the cathodic current under the CV test; the obtained charge values were divided by the composite mass of the working electrode. The electrode areal capacity (mAh cm$^{-2}$) was obtained considering only the cathodic surface directly faced to lithium metal, i.e., 2 cm$^2$ (1.6 cm diameter). Subsequently, a series of galvanostatic cycles run with the same charge current (C/5) at increasing discharge currents (C/5, C/3, and C/1) were performed to investigate how increasing currents could affect electrode performances. Finally, repeated charge/discharge cycles at C/1 to evaluate the cell stability under prolonged cycling of the different formulations were performed.

## 3. Results

Figure 1a reports the weight loss normalized to mass variation evaluated by TGA under Ar/O$_2$ of the LMNO cathodes with PU, PEDOT:PSS, and PVdF binders, and of the pristine LMNO powders, the characteristic decomposition temperatures are highlighted by the derivatives of the TGA profiles reported in Figure 1b. While the curve for LMNO-Powder indicates that no significant mass variation occurs, for the composite electrodes two minima exist. The first one is located below 400 °C and is related to the binder degradation. The second one, more evident, at around 500 °C is related to the combustion of the carbonaceous content (Super C45 or Super C65). All the binders degrade above 200 °C with pullulan being the less stable. PEDOT:PSS binder shows a more constant decomposition rate with the derivative minimum at 400 °C.

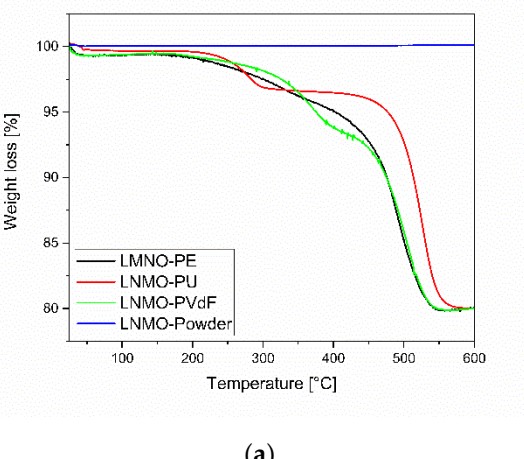
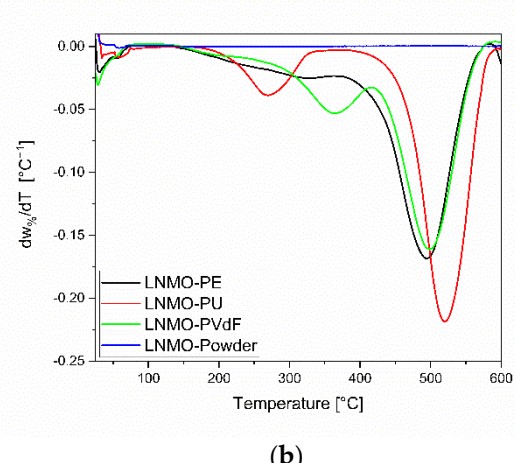

(**a**) 　　　　　　　　　　　　　　　　　　　　　　　(**b**)

**Figure 1.** (**a**) Weight loss normalized to mass variation and (**b**) Derivatives of the TGA profiles under Ar/O$_2$ of the three electrodes (the aluminum mass has been excluded) and of the pristine LMNO powder.

The XRD patterns of the pristine LMNO powder and of the LMNO-PU, LMNO-PE, and LMNO-PVdF electrodes are reported in Figure 2. The patterns overlap and no significant difference was observed in the XRD spectra after the water processing. This suggests that the bulk lattice of LMNO was well maintained after electrode production by PU or PEDOT:PSS.

At first, galvanostatic charge/discharge cycles at a C-rate of C/10 were performed to form a stable CEI and to evaluate the specific capacity of the tested formulations. Selected cell voltage profiles of the conditioning cycles are reported in Figure 3.

The cell voltage profiles reported in Figure 3a feature the plateau expected for LMNO electrodes. The discharge curve derivatives shown in Figure 3b enabled a better evaluation of the cell voltage plateau. The plateau at 3.97 V is related to the Mn$^{3+}$/Mn$^{4+}$ pair, the one at 4.67 V is related to Ni$^{2+}$/Ni$^{3+}$, and that at 4.72 V corresponds to the Ni$^{3+}$/Ni$^{4+}$ redox couples. [4]. The PEDOT:PSS water-based formulation, featured the highest specific LMNO discharge capacity (130 mAh g$^{-1}$), followed by LMNO-PVDF (124 mAh g$^{-1}$) and LMNO-PU (119 mAh g$^{-1}$). After the GCPL conditioning cycles at C/10, CVs at 50 μVs$^{-1}$ were carried out to highlight the presence of any secondary faradaic reactions related to

the different exploited binders. The voltammograms are reported in Figure 4. For all the tested cells, the CVs overlap and show well defined redox peaks that are related to the $Mn^{3+}/Mn^{4+}$, $Ni^{2+}/Ni^{3+}$ and $Ni^{3+}/Ni^{4+}$ redox pairs. No additional signals can be appreciated. The voltammetric test confirmed that LMNO-PE is outperforming in terms of specific capacity. It featured 130 mAh g$^{-1}$ with a coulombic efficiency of 99%. LMNO-PU exhibited 127 mAh g$^{-1}$ with a coulombic efficiency of 96.5%. For LMNO-PVdF the specific capacity and coulombic efficiency were 124 mAh g$^{-1}$ and 97.6%, respectively.

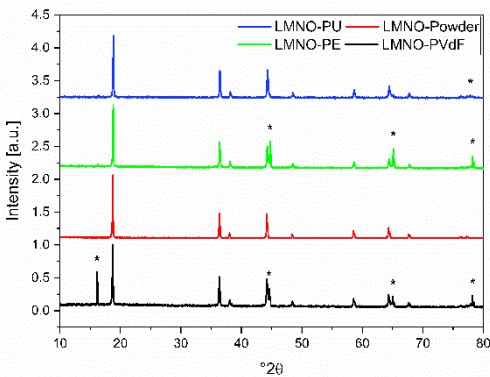

**Figure 2.** XRD patterns of the pristine LMNO powder and of the LMNO-PU, LMNO-PE, and LMNO-PVdF electrodes. (* Sample holder).

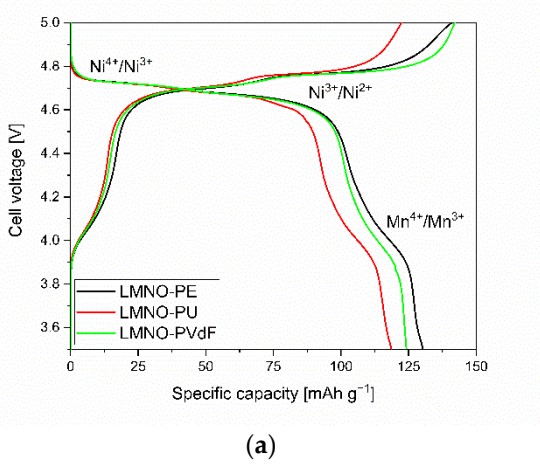

(**a**)

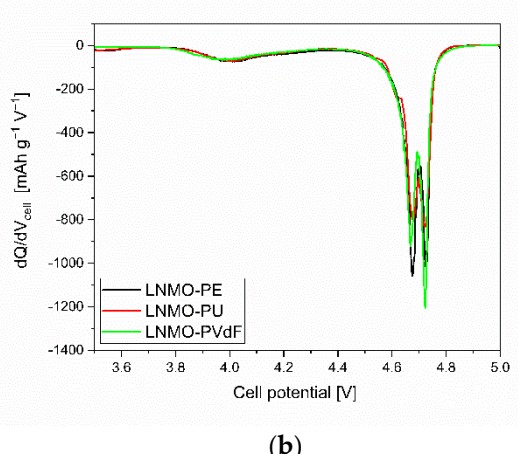

(**b**)

**Figure 3.** (**a**) Cell voltage profile vs. specific capacity and (**b**) derivative of the specific discharge capacity vs. cell voltage at C/10 of the different coin cells assembled with LMNO-PU, LMNO-PE, or LMNO-PVdF.

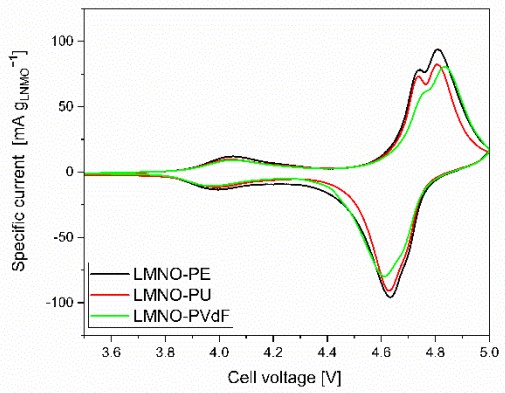

**Figure 4.** Cyclic voltammogram of the coin cells assembled with LMNO-PU, LMNO-PE, or LMNO-PVdF at 50 μVs$^{-1}$.

To investigate the response of the proposed electrodes formulations at high current, GCPL tests at increasing discharge current were carried out. In particular, the cells were charged with a C/5 current (calculated on the nominal capacity of LMNO) up to 5 V and discharged to 3.5 V at increasing C-rates, namely, C/5, C/3, and C/1. Figure 5 reports the trend of the specific capacity delivered at each cycle and current. All the cells featured a good rate capacity. At each C-rate, the highest specific capacity was achieved with LMNO-PE. Indeed, at C/5, LMNO-PE featured 136 mAh g$^{-1}$. This value decreased by 17% (to 113 mAh g$^{-1}$) when the current increased at C/1. For LMNO-PU and LPNMO-PVdF, the specific capacity at C/5 was 111 mAh g$^{-1}$ and 114 mAh g$^{-1}$, respectively, and it decreased to 103 mAh g$^{-1}$ at C/1. Overall, the CV and GCPL data demonstrate the feasibility of the use of water processed, high voltage cathodes based on the biopolymer pullulan or the electronically conducting PEDOT:PSS. Moreover, the good performance of the PEDOT:PSS-based formulation, suggests that the electronically conducting nature of the PEDOT:PSS binder, improves the electronic percolation network of the LMNO particles, and enables a better exploitation of the active materials even at high currents.

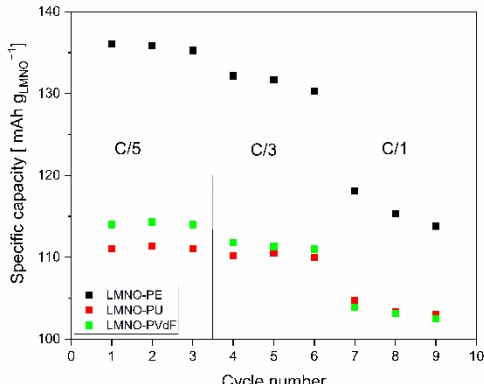

**Figure 5.** Specific capacity normalized to the LMNO content of the lithium metal cells with LMNO-PE, LMNO-PU, or LMNO-PVdF cathodes, under galvanostatic discharge at increasing current.

The GCPL at different C-rates have been analyzed to calculate the cell energy and power densitie reported in the Ragone plots in Figure 6. The cell energy ($E_{cell}$) has been calculated as the integral of the cell voltage under galvanostatic discharge conditions by introducing into the following equation the factor 3600, in order to convert the energy unit from Joule to Wh.

$$E_{cell} = i \int_0^\tau \frac{V}{3600} dt \tag{1}$$

where $i$ is the current, $V$ is the cell discharge voltage, and $\tau$ is the discharge time. From $E_{cell}$, the average cell power ($P_{cell}$) has been calculated according to the following equation.

$$P_{cell} = \frac{E_{cell}}{\tau} \tag{2}$$

For each tested cell, the specific energy ($E$) and the power density ($P$) have been calculated according to the following equation.

$$E = \frac{E_{cell}}{m_{cat} + m_{an}} \tag{3}$$

$$P = \frac{P_{cell}}{m_{cat} + m_{an}} \tag{4}$$

where $m_{cat}$ is the experimental mass of the composite cathode (see Table 1), and $m_{an}$ is the mass of the anode. The tested cathodes featured a capacity between 0.5 and 1 mAh,

while the anodic lithium foil was 300 μm thick. The anode theoretical capacity ($Q_{Li}$) can be calculated by Equation (5).

$$Q_{Li} = Q_{Li}^t \cdot \rho_{Li} \cdot A \cdot t \tag{5}$$

where $Q_{Li}^t$ is the theoretical lithium specific capacity (3860 mA g$^{-1}$), $\rho_{Li}$ is the lithium density (0.53 g cm$^{-3}$), A is the electrode area (2 cm$^2$), and t is the lithium thickness. According to Equation (5), a lithium anode with a thickness of 300 μm features a capacity of 120 mAh, which is c.a. 100 times the one of the prepared cathodes. Therefore, the mass of metallic lithium in a balanced cell can be significantly reduced. A metallic lithium foil of 15 μm with the same area would bring about 6 mAh, which is still from 6 to 15 times the capacity of the tested cathodes. Lithium foils featuring thickness smaller than 15 μm are extremely hard to handle. Indeed, reducing lithium foil thickness below 10 μm is one of the greatest challenges that metallic lithium batteries manufacturers will face in the years to come. A valuable solution is to produce thin lithium films on copper substrates [32].

The specific energy and power values reported in Figure 6, are projected data estimated for lithium metal cells assembled with the cathodes detailed in Table 1 and an optimized lithium foil of 15 μm. The highest values can be delivered by cells with LMNO-PE. The specific energy of the cells ranged from 420 Wh kg$^{-1}$ (at C/10) to 355 Wh kg$^{-1}$ (at C/1) for LMNO-PE, while the delivered power was between 40 Wh kg$^{-1}$ and 385 W kg$^{-1}$, respectively. In these evaluations, current collectors, separators, and electrolyte mass were not included. However, we would like to underline that these values should be considered just as preliminary. Indeed, extrapolation of the results achieved at lab-scale to project up-scaled battery performance is a very challenging task that should consider active, composite electrode and electrolyte compositions, composite electrode, current collector, and separator thickness, as well as overall cell size (cylindrical, pouch, or prismatic). In [33], an extremely useful tool for the prediction of full cell performance metrics starting from measurement results on the electrode level was proposed. The impact of "death components" on the gravimetric parameters largely depends on electrode and current collector thicknesses. The higher is the electrode thickness, the higher is the specific energy. In this work, we used thin electrodes; hence, the impact of the separator (1.6 mg m$^{-2}$) and electrolyte (200 μL) masses was not negligible.

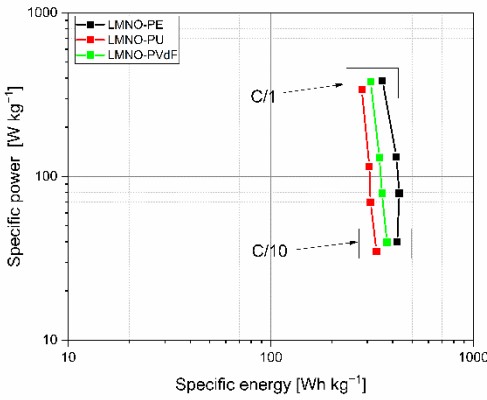

**Figure 6.** Ragone plot of the lithium metal coin cell s with LMNO-PE, LMNO-PU, or LMNO-PVdF cathodes.

Finally, cycling stability tests were performed at C/1, which can be considered a preliminary, accelerated condition. Worth noting, we used an as received lithium metal anode and no SEI forming additives were present in the electrolyte. Figure 7 reports the capacity retention ($C_{ret}$) over cycling, which was calculated according to Equation (6).

$$C_{ret} = Q(t)/Q_0 \tag{6}$$

where $Q(t)$ is the capacity delivered at the ith cycle, while $Q_0$ is the value exhibited at the first cycle. The cells assembled with the water-processed cathodes featured outstanding

cycle stability. Indeed, LMNO-PU performed like LMNO-PVdF and retained 80% of the initial capacity after 200 cycles. Outstanding is LMNO-PE that features an excellent capacity retention of 100%. This test should be taken as a preliminary one and should be validated by longer cycle-life tests. However, it already clearly demonstrates the better stability of PEDOT:PSS-based cathodes vs. LMNO-PVdF since the first cycles.

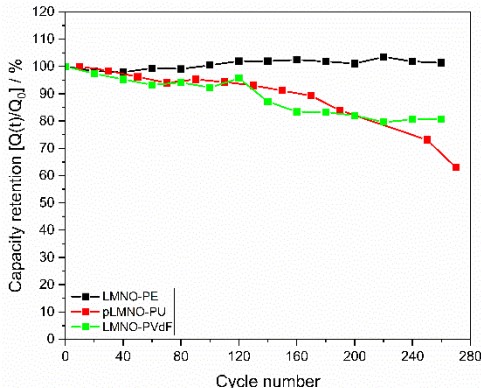

**Figure 7.** Capacity retention of the tested cell under cycling at C/1.

## 4. Conclusions

Overall, this paper demonstrates the feasibility of lithium metal battery cells assembled with high potential, water processed cathodes, and the LP30 electrolyte. LMNO electrodes were fabricated using two different water-processable binders: pullulan (PU) or the bifunctional electronically conductive PEDOT:PSS. LMNO was a commercial powder. The voltammetric and galvanostatic tests indicated that PU-based electrodes perform like the conventional PVdF-based ones, and represent a viable alternative to the latter binder. The use of PEDOT:PSS significantly improves specific capacity and capacity retention at different current rates and over cycling. At C/10, LMNO-PE featured 130 mAh g$^{-1}$, which is even slightly higher than the nominal value reported by the LMNO powder provider (125 mAh g$^{-1}$). The most interesting result is that lithium metal cells assembled with PEDOT:PSS-based cathode featured an outstanding capacity retention of 100% over 200 cycles carried out at C/1 and with a high cut-off voltage of 5 V. These results are even more important, considering that LMNO-PE is produced in ambient conditions (non-controlled condition), without using toxic solvents. The good cycling stability of LMNO-PE could be related to the ability of PEDOT:PSS to be a "barrier" against undesired side reactions of LMNO with the electrolyte. On the other hand, this effect has already been observed for cathodes produced with PVdF binder and PEDOT:PSS-coated LMNO particles [19–23]. Hence, this work demonstrates that it is possible to produce LMNO cathodes with aqueous binders maintaining the same performance of PVdF, in the case of pullulan, and improving them by the direct use of an electronically conductive binder such as PEDOT:PSS, without the need of LMNO particle coating. Comparison of PU and PEDOT:PSS-LMNO cathodes demonstrates that, besides the possibility of designing sustainable manufacturing, functional polymers with inherent electronic conductivity, play a key role, enabling specific energy and power performances greater than those of PVdF based ones. In addition, our study aims at widening the use of PU and PEDOT:PSS, which are considered key materials in the packaging industry and organic electronics, in an additional, strategic technological sector, namely, LIB manufacturing.

**Author Contributions:** Conceptualization, F.S.; methodology, F.S.; validation, F.S.; formal analysis, A.B., F.P., E.P., F.M. and F.S.; investigation, F.M., A.B. and E.P.; resources, F.S.; data curation F.M., A.B., F.P. and E.P; writing—original draft preparation, A.B., F.P., E.P. and F.S.; writing—review and editing, F.S., F.P, A.B. and E.P; visualization, A.B.; supervision, F.S.; project administration, F.S.; funding acquisition, F.S. All authors have read and agreed to the published version of the manuscript.

**Funding:** This research was funded by "Piano Triennale di Realizzazione 2019–2021" Accordo di Programma Ministero dello Sviluppo Economico (MISE)- Agenzia nazionale per le nuove tecnologie, l'energia e lo sviluppo economico sostenibile (ENEA) by "New generation of electrochemical energy storage systems" Progetto Alte Competenze per la Ricerca e il Trasferimento Tecnologico POR FSE 2014/2020-REGIONE EMILIA ROMAGNA and by "Programma Regionale attivita' produttive 2012–2015 (Attivita' 5.1, Sostegno allo sviluppo delle infrastrutture per la competitività e per il territorio -REGIONE EMILIA ROMAGNA).

**Conflicts of Interest:** The authors declare no conflict of interest. The funders had no role in the design of the study; in the collection, analyses, or interpretation of data; in the writing of the manuscript, or in the decision to publish the results.

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
