# Peer review of "Performance Comparison of LMNO Cathodes Produced with Pullulan or PEDOT:PSS Water-Processable Binders"

_energies, doi:10.3390/en15072608_

Round 1

Reviewer 1 Report

Dear Authors,

please find some minor comments enclosed below.

Sincerely, 

the Reviewer.

Comment 1

Lines 34-38, please add a reference (i.e. https://ec.europa.eu/energy/sites/ener/files/documents/batteries_europe_strategic_research_agenda_december_2020__1.pdf)

Comment 2

Line 41, explicit/define EBA.

Furthermore, would be preferable to add a citation to support “…EBA goals…”

Comment 3

Lines 45-46, check the language style.

Comment 4

Line 46, “NMP has been listed by Europe as a high concern substance.” please add a reference.

Comment 5

Line 56, “…for its high highly water-solubility…” check the language style.

Comment 6

Lines 58-60, “PU has been reported as a binder component of silicon anodes, as well as Pullulan:glycerol with 1:1 mixture (PU) exhibited excellent binding capability for thick carbon electrodes of supercapacitors and Li(Ni0.5Mn0.3Co0.2)O2 (NMC532) cathodes [7, 10-12].”

The acronym PU is defined after its first use, check and correct.

Comment 7

Lines 73-74, “…have a beneficial effect at high current densities, reaching 81 mAh g-1 at 1600 mA g-1 [15].”, please try to be more clear.

Comment 8

Figure 2, the quality of the image (as well as of the other Figures present in the manuscript) is low, please try to improve it by choosing a different type of file format.

Furthermore these results could be better represented. Specifically, emphasizing peaks around 200 ºC by the change of scale in the ordinate axis.

Comment 9

Figure 3. Despite of the poor quality of the image, comparing curves the plateau addressed to the Ni3+/Ni2+ redox couple is clearly distinguishable for the LMNO-PU curve, while is barely distinguishable in LMNO-PVdF and almost undistinguishable in LMNO-PE. At the same time, the occurrence of such plateau correspond to a decrease of the specific capacity. Please discuss this evidence according to the cited reference [2] (10.1039/D0TA02812F).

Comment 10

Line 160, the acronym GCPL is defined only in line 339 (Materials and methods section).

The same for CV (Line 183).

Maybe the choice of having the materials and methods in the end of the manuscript (Section 4) doesn’t help preventing these problems along the manuscript reading.

Comment 11

Following the previous comment, check the numeration of the Materials and methods subsection at the Line 337.

Comment 12

Lines 195-197, “The cell energy (Ecell) has been calculated as the integral of the cell voltage under galvanostatic discharge condition (Wh) by introducing in the following equation the factor 3600”.

Why? Please clarify

Line 201, “…where V is the cell discharge voltage, is the discharge time.” Something is missing, check and correct.

Line 211 “The anode capacity of the anode…”, check the language style.

Reviewer 2 Report

The present study investigated LiNi0.5Mn1.5O4 (LMNO) electrodes fabricated using water-processable binders pullulan and PEDOT:PSS. It was found that at C/10, the specific capacity of LMNO-PEDOT:PSS and LMNO-PU were respectively 130 mAh g-1 and 127 mAh g-1, slightly higher than that of LMNO-PVdF. In lithium metal cell, PEDOT:PSS cathode delivered capacity retention of 100% over 200 cycles at C/1 rate with a cut-off voltage of 5 V. Specific comments are given below.

  1. Although the significance of exploring pullulan and PEDOT:PSS binders have been demonstrated individually, why are these two binders chosen to be compared in this study? They are vastly different in structure, so what conclusion could be obtained from this comparison? What are the key significance and advances of the present study?
  2. What is purpose of Fig 1? This is a highly common protocol and there does not seem to be necessity to include a photo here.
  3. For Fig. 2, the authors claimed that PEDOT:PSS shows an excellent thermal stability with a minimum and constant mass loss up to 400 C. This conclusion does not seem to be well supported. The curve only indicates more constant decomposition rates, rather than supreme thermal stability. Thermal stability should be evaluated with a combination of overall weight loss and post-heating electrode performance variation.
  4. For rate test (Fig 6), commonly reversibility would be demonstrated, namely to resume to C/5 by the end of the test. Why is such data not included?
  5. For cycling stability tests, only 160-200 cycles are too short to be fully credible.
  6. The authors need to carefully proofread the manuscript. For example, the beginning of Section 2 has one remaining paragraph from template file.

Reviewer 3 Report

The paper may be published after minor revisions:

1) Some discussion on the mechanical properties of the PEDOT:PSS binder should be added;

2) There are some template phrases left in the manuscript which should be removed:

Results , line 111-112
This section may be divided by subheadings. It should provide a concise and precise description of the experimental results, their interpretation, as well as the experimental conclusions that can be drawn."

Reviewer 4 Report

The paper compares three different binders for LMNO high voltage spinel cathodes. PEDOT:PSS and pullulan are two water-processable binders that can be used to replace commercial PVDF binder, which is more environmentally friendly. Also, it is found that PEDOT:PSS binder improves the cycle life of the LMNO cells. Overall, the paper is straightforward and provides sufficient data to support the finding. However, the following comments need to be addressed.

  1. Chemical vapor deposition polymerization is an important technique to in-situ grows PEDOT on cathode materials. The application of such a technique needs to be introduced in the introduction when introducing the synthesis of PEDOT and the effect of PEDOT coatings on cathode materials performance. (https://doi.org/10.1021/acsami.8b08711, https://doi.org/10.1021/acsami.0c20978).
  2. Figure 4: charge curves should also be shown here to compare the overpotential between charge curves and discharge curves.
  3. Line 201: a term is missing when introducing “is the discharge time”. Also, other terms, such as i and Ecell, should also be introduced.
  4. Line 203: Similarly, Pcell and t need to be introduced
  5. Equation (3) and (4) only considers the mass of cathode and anode. How about other components, such as separator and electrolytes?
  6. Line 221: Actually, it is already very hard to handle 50 um thick Lithium foil. Please provide a reference for the 15 um Li foil.
  7. A paper should have a Conclusion section.
